# Dietary Phytochemicals in Zinc Homeostasis: A Strategy for Prostate Cancer Management

**DOI:** 10.3390/nu13061867

**Published:** 2021-05-30

**Authors:** Chandra K. Singh, Gagan Chhabra, Arth Patel, Hao Chang, Nihal Ahmad

**Affiliations:** 1Department of Dermatology, University of Wisconsin, Madison, WI 53705, USA; csingh@dermatology.wisc.edu (C.K.S.); gchhabra@dermatology.wisc.edu (G.C.); ajpatel6@wisc.edu (A.P.); hchang@dermatology.wisc.edu (H.C.); 2William S. Middleton VA Medical Center, Madison, WI 53705, USA

**Keywords:** prostate malignancy, zinc transporters, metallothioneins, natural agents

## Abstract

Studies have suggested an important role of the trace element zinc (Zn) in prostate biology and functions. Zn has been shown to exist in very high concentrations in the healthy prostate and is important for several prostatic functions. In prostate cancer (PCa), Zn levels are significantly decreased and inversely correlated with disease progression. Ideally, restoration of adequate Zn levels in premalignant/malignant prostate cells could abort prostate malignancy. However, studies have shown that Zn supplementation is not an efficient way to significantly increase Zn concentrations in PCa. Based on a limited number of investigations, the reason for the lower levels of Zn in PCa is believed to be the dysregulation of Zn transporters (especially ZIP and ZnT family of proteins), metallothioneins (for storing and releasing Zn), and their regulators (e.g., Zn finger transcription factor RREB1). Interestingly, the level of Zn in cells has been shown to be modulated by naturally occurring dietary phytochemicals. In this review, we discussed the effect of selected phytochemicals (quercetin, resveratrol, epigallocatechin-3-gallate and curcumin) on Zn functioning and proposes that Zn in combination with specific dietary phytochemicals may lead to enhanced Zn bioaccumulation in the prostate, and therefore, may inhibit PCa.

## 1. Introduction

Prostate cancer (PCa) is the second most diagnosed form of cancer in the United States among men. Around 248,530 new PCa cases and 34,130 PCa-related deaths are expected to occur in 2021 in the United States [1]. The average age of PCa diagnosis is 66, with approximately 6 in 10 men being diagnosed aged above 65 years old. Given the prevalence of the disease and the economic burden to manage it, finding a treatment for PCa is a priority among researchers currently.

The prostate gland is made up of peripheral, central and transition zones, with the peripheral zone being the largest and the most common site for prostate malignancy. Studies have shown that prostate cells accumulate high levels of Zn, resulting in inhibition of mitochondrial aconitase, and truncation of Krebs cycle at the first step of citrate oxidation, causing citrate to be the final product. Inhibition of mitochondrial aconitase and citrate oxidation is unique to prostate epithelial cells, as it is lethal in other mammalian cells [2]. Interestingly, the amount of Zn present in cancerous prostate is significantly lower (by 60–80%) than that in a healthy prostate (reviewed in [3,4]). In the absence of high Zn levels, mitochondrial aconitase is not inhibited and citrate oxidation proceeds in the Krebs cycle. This makes malignant cells more bioenergetically efficient than normal prostate cells (reviewed in [3,4]). This also suggests that Zn is vital in the maintenance of a healthy prostate. The depleted concentrations of Zn have been shown to be related to prostate malignancy, and therefore, adequate Zn levels if able to reach the prostate may halt PCa progression. Studies have shown that Zn supplementation is an inefficient way to supplement the required high Zn concentrations in the prostate due to malfunctioning of regulators of Zn homeostasis [5,6]. Interestingly, certain dietary phytochemicals have been shown to affect Zn levels by modulating Zn transporters and their regulators. In this review, we discussed the role of Zn and the effects of selected dietary phytochemicals on Zn functioning. Based on the studies available, we also propose that Zn in combination with specific dietary phytochemicals may lead to enhanced Zn bioaccumulation in the prostate, and therefore, may inhibit PCa.

## 2. Biochemistry of Zn in the Healthy Prostate

Prostatic fluid and the peripheral zone of the prostate gland, which make up about 70% of the human prostate gland, contain extremely high levels of Zn and citrate [3,7,8]. The concentration of Zn in the prostatic fluid is about 500 times greater (which has been estimated as 590 µg/g wet weight) than that in blood plasma while that in the prostate peripheral zone is about 10–20 times greater than that in similar body tissues [4]. These extraordinarily high concentrations are because peripheral zonal prostate epithelial cells are Zn-accumulating cells, which are accountable for the production and secretion of citrate from the prostate [3,8]. Mitochondrial aconitase, which catalyzes the isomerization of citrate to produce isocitrate in an early step of the Krebs cycle, is greatly inhibited in Zn-accumulating prostate cells [9,10,11]. This leads to inhibition of citrate oxidation causing high citrate levels in prostatic fluid, which is an important constituent of semen [10,11]. The presence of Zn in these epithelial cells directly inhibits the reaction, holding the citrate/isocitrate ratio in the relevant prostate tissue to 30–40 nmols/g, which is far greater than 9–10 nmols/g found in other cells (reviewed in [3]). In other words, high concentrations of citrate and high citrate/isocitrate ratios in the epithelial cells of the prostate’s peripheral zone are caused by high concentrations of Zn in these cells. As a consequence, high Zn interrupts the oxidation of citrate and only 14 adenosine triphosphate (ATP) per glucose molecule is produced as opposed to the typical 38 ATP that comes from the complete oxidation of glucose [3,9]. These processes highlight the extremely important functions of Zn in the healthy prostate.

## 3. Zn and Metabolic Reprogramming in PCa

Many cancerous tumors encounter the Warburg effect, which describes the tendency of malignant cells to favor aerobic glycolysis over oxidative phosphorylation in regards to ATP production [12]. This allows tumors to increase their biomass and avoid potential mitochondrial stress, which would inhibit cell division. PCa, in contrast, does not exhibit the Warburg effect until in the late stage and after many mutations. Healthy prostate cells are capable of avoiding oxidative phosphorylation due to Zn’s unique metabolic effects, as discussed above. Thus, early-stage PCa cells do not increase glucose uptake. Because malignant prostate cells have decreased levels of Zn, the oxidation of citrate is no longer efficiently inhibited [13,14]. The produced isocitrate from the oxidation reaction is then capable of entering the Krebs cycle, increasing the ATP produced to meet the energy consumption requirements of rapidly growing and reproducing malignant cells [3,5,8,11,12]. Later stages of PCa, however, do favor aerobic glycolysis. This preference leads to higher intracellular levels of lactic acid, which is a cytotoxin. In order to prevent damage, malignant cells express monocarboxylate transporters that carry molecules having one carboxylate group across biological membranes, to reduce the lactic acid content in the cells [12].

Lipids are another essential source of energy for PCa progression and androgens play a major role in stimulating lipogenesis [15,16]. PCa cells express androgen receptors (AR), a ligand-dependent transcription factor, to increase levels of lipid synthesis markers such as fatty acid synthase, sterol regulatory element-binding protein 1, and steroyl CoA desaturase via androgen signaling [16]. AR signaling has been demonstrated in all stages of PCa, and one of the downstream targets of AR, prostate-specific antigen (PSA), has been established as a clinical biomarker for PCa diagnosis, prognosis, and progression (reviewed in [17]). Interestingly, Zn is known to inhibit AR and PSA signaling to suppress the growth of PCa cells [18]. The effect of Zn in androgen-independent and dependent PCa is illustrated in Figure 1. Androgen deprivation therapy, which lowers the levels of androgens in the prostate, thereby lowering the potential for cancer cells to accumulate the energy needed to grow, divide, and invade, has been accepted as the first-line treatment of symptomatic metastatic PCa [19]. A critical point in the progression of prostate malignancy is when cells begin to utilize lipid synthesis rather than relying on androgen regulation. This change is crucial because it creates a disease that does not respond to androgen deprivation therapy. This development is known as castration-resistant PCa, which is determined by a rising PSA in an environment where androgens are castrated [19].

The expression of glucose transporter types 1 and 2 (GLUT1 and GLUT2) are decreased in androgen-unresponsive PCa [23]. Phosphofructokinase 1 (PFK1), a regulatory enzyme that plays a rate-limiting step of glycolysis, was found to be markedly increased in androgen-responsive cells but decreased in androgen-unresponsive cells in comparison to healthy prostate cells. Lactate dehydrogenase (LDH), which facilitates the production of lactate from the metabolism of glucose, has been found to have increased enzymatic activity in androgen-unresponsive cells, though there is no significant difference in the LDH protein levels between androgen-unresponsive and androgen-responsive cells [23]. The progression of PCa and its characterization of androgen unresponsiveness can thus be linked to changes in glucose uptake via GLUT1 and 2 levels, PFK1 expression, and LDH enzymatic activities. All these molecular changes suggest the role of metabolic reprogramming in PCa progression ultimately leading to castration-resistant PCa.

## 4. Role of Zn in the Healthy Prostate and PCa

Maintaining optimal intracellular Zn levels is crucial for many cellular functions, especially because Zn serves as a catalytic or structural cofactor for a variety of different proteins. Next only to iron, Zn has the second-highest concentration in the body among all trace elements. Since it is a trace element, its presence in the body depends on the dietary habits and physiological factors of the individuals. A systematic review of 105 studies analyzing Zn content in the human prostate has found that Zn levels also depend on age, androgens level, prostatic region, as well as on fraction of prostate tissue studied and analytical method used in the study [24]. Zn is a fundamental component in the body’s defense mechanisms, as it is involved in mitosis, healing body wounds, breakdown of carbohydrates and the functioning of the immune system [25]. Zn plays a role in maintaining protein structure, as well as the regulation of gene expression. Around 100 different enzymes require the presence of Zn to function, including carbonic anhydrase, RNA polymerase, alcohol dehydrogenase, and the hormone insulin. It has been estimated that ~3000 genes of the human genome encode Zn proteins [26] (reviewed in [27]). Overall, Zn has been shown to be essential for ~300 different cellular processes [28]. The prostate is heavily dependent on Zn in order to function properly. Zn inhibits terminal oxidation in the mitochondria, thereby regulating prostate energy production, and cell division overall [3]. Studies have shown that lack of Zn can result in increased DNA breakage, as well as overall DNA damage, suggesting the role of Zn in regulating DNA repair. Zn also functions as an antioxidant and prevents oxidative stress, a hallmark of many chronic diseases. Its absence results in the accumulation of free radicals in the prostate [27,28].

Interestingly, it has been found that Zn levels in the prostate decrease as PCa progresses suggesting that Zn levels influence the ability of PCa to progress [29]. PCa patients have shown increased levels of Zn in urine as well. Low levels of Zn precede the occurrence of low citrate levels, which are always correlated with oncoming malignancy. In fact, levels of Zn and citrate can be used to determine PCa stages. It has been established that malignant PCa almost never contains high levels of Zn. Therefore, low Zn levels can be used as a strong indicator of malignant PCa, or a “pre-malignant” stage. A recently published meta-analysis established a possible correlation between serum Zn levels and the stages of PCa [30]. After analyzing 14 different studies, the trend established was that lower serum Zn levels were found in PCa patients when compared to benign and control patients. On the contrary, patients with benign tumors were found to have higher levels of Zn. This correlation could potentially be used as an indicator for the progression of PCa [30]. There is also a link between Zn concentrations in the prostate and levels of certain Zn transporters, which are important in the uptake and accumulation of Zn in prostate cells [31]. This correlation suggests that the progression of PCa affects Zn transporters in the prostate or vice-versa. Thus, maintaining an optimal intracellular Zn concentration by modulating Zn transporters in the prostate could be key to find a way to manage PCa.

## 5. Lower Levels of Zn Support Prostate Malignancy

Because the prostate peripheral zone’s epithelial cells evolved as Zn-accumulators, they must also have developed characteristics to prevent high Zn concentrations induced cytotoxicity in the healthy prostate. However, the development of prostate malignancies involves the reduction of typical Zn-accumulating abilities and so it can be deduced that prostate malignancies are vulnerable to Zn cytotoxicity. This is supported by reports demonstrating that malignant prostatic cells experience cell proliferation inhibition, apoptosis, and decreased ability to migrate and invade when Zn levels are increased [32]. Therefore, cancerous prostate cells have evolved several methods to reduce Zn concentrations including dysregulation of expression of multiple Zn transporters (ZIPs and ZnTs; also known as solute carrier family 39 members and solute carrier family 30 members, respectively), an increase in Zn importer negative regulators Ras Responsive Element Binding Protein 1 (RREB1), upregulation of microRNAs with similar consequences (microRNA-96, microRNA-182 and microRNA-183), and overexpression of DNA-binding transcription factor Homeobox B13 (HOXB13) [33,34,35,36,37]. Activation of RREB1 has been found to downregulate Zn importer ZIP1 in PCa by binding to its promoter region [35]. Similarly, overexpression of miR-183-96-182 individually or in a cluster has been demonstrated to suppress mRNA levels of six Zn transporters (ZIP1, ZIP3, ZIP7, ZIP9, ZnT1, and ZnT7) and reduce Zn uptake [36]. HOXB13 is known to promote PCa progression and activate nuclear factor kappa B (NF-κB) via upregulation of Zn exporter ZnT4 resulting in decreased intracellular Zn levels in PCa [37]. Here, it is important to mention that activation of transcription factor NF-κB is one of the key pathways in the regulation of proangiogenic and pro-metastatic molecules in PCa pathogenesis.

As an example demonstrating the benefit to malignant prostatic cells for having lower Zn levels: Zn’s inhibition of PCa cell growth involves an increase of cyclin-dependent kinase inhibitor 1a (p21) gene expression [34,38]. It has been found that p21 is decreased in the nuclei of malignant Zn-deficient PCa cells [20]. Similarly, treatment with physiological levels of Zn prevents NF-κB from continuing its activities in PCa cells [39]. Zn-deficient malignant prostate cells use NF-κB pathway to express pro-tumorigenic cytokines [22]. Zn deficiency allows an increase in hypoxia-inducible factor 1α (HIF-1α) levels, which is associated with increased invasiveness of C27 PCa cells, along with an increase in vascular endothelial growth factor, interleukin-6, interleukin-8, and matrix metalloproteinase, which all have been shown to be pro-metastatic and proangiogenic [40]. In an *in vitro* experiment, treatment with Zn significantly reduced the expression of pro-inflammatory cytokines known to promote PCa. Zn treatment lowers the levels of molecules that tumor cells use to promote new blood vessel formation to feed the growing tumor, as well as proteins that promote invasion and metastatic spread cytokines [22,34,40]. Zn also plays a role in the regulation of apoptosis in PCa cells by releasing cytochrome c, which activates caspase-9 and -3, and poly(ADP-ribose) polymerase (PARP) cleavage [41]. Thus, a decrease in Zn levels results in a decrease of apoptosis in malignant cells, which allows the proliferation of PCa cells and support prostate malignancy.

## 6. Important Regulators of Zn Homeostasis in the Prostate

The key components of maintaining Zn homeostasis in the human prostate are Zn importer proteins ZIPs (Zrt- and Irt-like Proteins, also known as solute carrier family 39 members (SLC39s), Zn exporter proteins ZnTs (Zn Transporters, also known as solute carrier family 30 members (SLC30s), and Zn proteins with storing and releasing mechanisms (metallothioneins, MTs) [5,42]. ZIPs transport Zn into the cell from the extracellular environment and these actions are followed by ZnTs transporting the newly imported intracellular Zn into organelles and/or back into the extracellular environment as needed. These transporters work in opposite directions to mediate optimal intracellular Zn levels, disturbance of which will lead to an increased chance of developing ailments including PCa. Meanwhile, MTs store free intracellular Zn ions in order to lower their concentration and release it as needed for essential processes [42]. All these mechanisms are interrupted during prostate malignancy, resulting in reduced intracellular Zn concentrations.

ZIP transporters also called ZIP importers are 14 members of family protein (ZIP1-14) generally transport Zn from the extracellular fluid into the prostate’s cells. ZIP1 is the most essential importer to move Zn into a cell and is typically found on basolateral membranes of prostatic cells. An overexpression of ZIP1 in RWPE-2 PCa cells decreased proliferation and induced apoptosis [43]. The downregulation of ZIP1 has been established as one of the primary causes of decreased Zn levels in prostate malignancies and a crucial early event in PCa development. ZIP2 and ZIP3 also play roles in the transportation of Zn, though to a lesser degree than ZIP1. Found on the apical surface of prostate cells, ZIP2 and ZIP3 are theorized to aid in the reuptake of Zn from the prostatic fluid. ZIP9 is the only member of ZIP family with membrane androgen receptor (mAR) characteristics [44]. A study in 2014, elucidated the intermediary role of ZIP9 in causing human breast and PCa, as it induced apoptosis in the presence of testosterone in breast and PCa cells. Unlike ZIP1, 2 and 3, ZIP9 mRNA expression was increased in human prostate and breast malignant tissues, which probably was because cells that divide rapidly require more Zn [45]. The role of other ZIP transporters is still under investigation, and their role is not very clear in PCa development and progression.

On the other hand, ZnT transporters also called ZnT exporters are 10 members of family protein (ZnT1-10) generally move intracellular Zn to organelles or extracellular environments [46]. These ZnT exporters, found in cell membranes as well as in the membranes of cellular organelles immediately bind to Zn ions as they enter the cell membrane, effectively eliminating the possibility of a large buildup or pool of Zn ions. ZnT1, found on the basolateral membrane of epithelial cells, is the only ZnT family transporter that exports Zn from cytosol to the outside of a cell, lowering cellular Zn levels [47,48]. ZnT4 is overexpressed in PCa tissues compared to healthy tissues. However, there is a lower level of ZnT4 expression in more advanced cases of PCa compared to earlier stages. Null mutations of ZnT7 have been found to promote the development of prostate tumors in mice, suggesting that the inactivity of ZnT7 could lead to PCa development. Earlier, we showed that there are significantly higher levels of mRNA expression of ZnT1, ZnT9 and ZnT10, and lower levels of ZnT5 and ZnT6 in human PCa compared to benign prostate tissues [49]. Further, the differential expression of these Zn exporters was varied with tumor stage and grade mostly in the early events of PCa development and progression. Additionally, these Zn exporters were predicted to interact with key tumor suppressors and promoter proteins known to play role in PCa pathogenesis [49].

MTs act as intracellular Zn reservoirs, releasing them for essential mechanisms and holding them to avoid cytotoxicity and maintain proper Zn homeostasis [42]. MTs are small proteins (~6 kDa), with large amounts of cysteine and their expression is increased by high levels of Zn [50,51]. MTs bind seven Zn ions to their cysteine groups, which is how they act as reservoirs of Zn. MTs levels have been found to be increased with age. The prostate is one of the tissues in the human body with high metallothionein 1 and 2 (MT1 and MT2) expression and has been found to be correlated with PCa progression. In human PCa cells, Zn was found to upregulate MT1 and MT2 [52]. MT1 and MT2 search for and collect extracellular Zn but do not compete with essential Zn-requiring proteins [42]. MT1 and MT2 transcription is regulated by metal response element-binding transcription factor-1 (MTF1). MTF1 induces the transcription of MTs when Zn ion concentrations are too high in the cytosol [52,53]. The resulting MT1 and MT2 proteins bind Zn, effectively lowering Zn’s cellular concentration and achieving homeostasis.

## 7. Challenges with Zn Supplementation

Approximately 90% of the Zn in the prostate are tightly bound to immobile macromolecules (protein-bound) and ~10% are loosely bound to mobile low molecular weight ligands such as Zn amino acids, Zn citrate and Zn metallothionein [9,32,54]. Very little of the exchangeable Zn is free in solution. Zn transporters can take free and mobile ligand-bound Zn only [55,56]. Over-the-counter Zn supplements (used in hopes to increase prostatic Zn concentrations to fight and/or prevent PCa) typically contain similarly bound Zn. Oral Zn supplementation has been found to significantly increase intra-prostatic Zn concentrations in rats [57]. Further, Zn supplementation in humans has been found to increase metallothioneins, which regulate the storage and release mechanisms of Zn [58]. Zn gluconate is a recommended chemical compound for the treatment of Zn deficiencies because it was found to have the most bioavailability in the prostate [59]. Zn sulfate is commonly used in supplementation in humans, but in a rat prostate, it has been found with the lowest prostate bioavailability compared to Zn gluconate and Zn citrate [59].

Though one might assume that Zn supplements would combat Zn deficiencies in PCa, there have been conflicting outcomes when it comes to the prevention and treatment of PCa using Zn supplements [60]. Some studies have found that Zn supplements decrease the risk of developing advanced stages of PCa [61] but others have found no significant effect at all [62], or even an increased risk of PCa [63]. There are several explanations for these conflicting data. One such possibility involves adverse metabolic effects, immune dysfunction and impaired antioxidant defense, which encourage the growth of cancers [64]. Some over-the-counter Zn supplementation options have been found to contain potentially harmful elements, such as cadmium (a potential prostate carcinogen), which could explain the discrepancies in studies [59]. The most probable explanation may be that even though Zn supplements increase the bioavailability of Zn in the body as a whole, they may not necessarily raise intra-prostatic Zn to the desirable levels due to dysregulation in Zn transporters in PCa. This was also evident from a study where direct intra-tumoral injection of Zn halted tumor growth in a xenograft model of PCa [65].

Several drugs/agents are being studied in conjunction with Zn to treat PCa. Zn has been found to sensitize PCa cells to Sorafenib, a multi-kinase inhibitor of proto-oncogenes rat sarcoma virus (RAS)/rapidly accelerated fibrosarcoma (RAF) kinases, the vascular endothelial growth factor (VEGF), and platelet-derived growth factor receptor (PDGFR) to induce apoptosis in PCa cells [66]. Interestingly, Livin, a member of the inhibitor of apoptosis proteins (unlike Survivin) was found to increase Zn-mediated apoptosis in PCa cells. Though the authors of this study conclude the synergistic effects of Zn and sorafenib against PCa, one can speculate that this combination may not be long-lasting as an increased level of Livin is already known to contribute to PCa cell proliferation [66]. This suggests that a combination of Zn with a drug/agent of interest should be evaluated carefully for optimal efficacy against PCa.

## 8. Zn Deficiency and PCa Epidemiology

Zn deficiencies can be found in all age groups of people, especially the elderly group [67]. It has been estimated that 17.3% of the world’s population has inadequate Zn ingestion from the diet [68]. Epidemiologically, conflicting data exist regarding Zn deficiency and PCa incidences. One study has suggested an inversed association between soil Zn concentrations linked with elevated groundwater use and PCa incidence in South Carolina, USA [69]. Despite South/South-East Asia being the most at-risk Asian region for Zn deficiency, Western Asia is found to be by far the Asian region with the highest PCa incidence rate. Even though Oceania has a low incidence rate of Zn deficiency, it has been found that it has the highest PCa incidence rate of all regions [70]. Similarly, although North America and Europe have low Zn deficiency incidences, North America and Western and Northern Europe have been found to have the highest PCa incidence rate in the world next to Oceania [70]. However, it has also been found that these rates are unreliable measures of the real rates of PCa incidence due to varied screening numbers. This oddity may also be explained by overdiagnosis; studies show that up to 42% of PCa cases in the USA and Europe could be due to overdiagnosis. Additionally, it is very well known that several other factors such as genetics and diet contribute to PCa incidences as well.

In the USA, African American men are disproportionately affected by PCa, as they have the highest PCa incidence rate (reviewed in [70,71]). Surveillance, Epidemiology and End Results (SEER) data show a higher PCa incidence in African Americans compared to European Americans in terms of age of onset, morbidity, and presentation with advanced cancer [72]. Even after adjusting for demographic, socioeconomic, clinical, and pathologic factors, the risk for PCa remained statistically higher for African American men [73,74]. Surprisingly, African countries have much lower rates of PCa incidence. It has been found that PCa diagnosis is as much as 40 times as prevalent for African Americans than for native Africans. This discrepancy could also be a result of differences in the prevalence of testing across nations. For example, early detection testing for PCa is not common in several African countries. One plausible explanation for this discrepancy may also be that Africa is a mineral-rich continent, and ground Zn concentrations in soil and water are high [75]; therefore, the African population may have natural adaption for low Zn accumulation (to escape Zn toxicity) by adjusting the levels of Zn-importers (ZIPs), Zn-exporters (ZnTs), or both. Recent studies suggest that the unusually lower concentration of Zn in PCa, driven by dysregulated Zn transporter system [11,13,14] could also be a reason for the observed PCa racial disparity, with a higher risk of this cancer in African Americans. One study has found downregulation of ZIP1 and ZIP2 in the normal prostate as well as in PCa tissues from African Americans when compared with age-matched European Americans [76]. In a recent study from our lab, we found significant differences in the expression levels of several Zn-exporters (ZnTs) in the PCa tissue samples of European Americans versus African Americans [49]. This information provides evidence that the racial disparities in PCa between African Americans and European Americans could be due to differences in the levels of Zn transporter protein expression in PCa tissues of the two populations, which may lead to decreased Zn levels within African American prostate cells and, thus, increased risk of developing PCa.

## 9. Naturally Occurring Dietary Phytochemicals Known to Enhance the Absorption/Bioaccumulation of Zn

Recent studies have found that bioavailable levels of Zn are affected in response to certain naturally occurring dietary phytochemicals. The effects of green tea, red wine and grape juice, were examined on Zn uptake in Caco-2 cells [77]. It has been found that these drinks (which are rich in polyphenols), along with tannic acid and quercetin, enhanced Zn uptake due to binding to Zn in competition with zinquin within these cells. Zinquin is a Zn selective fluorophore generally used to detect intracellular Zn levels in experimental models. Caco-2 is an immortalized human colorectal cancer cell line generally used as an *in vitro* experimental model of the intestinal epithelial barrier, and also in the pharmaceutical industry to predict the absorption of the drugs [78]. It is worth noting, however, that alcohol significantly impairs Zn uptake in humans [79]. These treatments (green tea, red wine, grape juice, tannic acid and quercetin) also increased the expression of metallothioneins indicating a higher level of Zn within the cells [77]. Similarly, food proteins containing amino acid residues including cysteine, histidine, serine, aspartate, and glutamate may also prove to be useful in maintaining proper Zn level due to their Zn chelating abilities, which create soluble Zn complexes resulting in increased Zn absorption [80,81]. Further, it has been found that adding pyridoxine (vitamin B6), a cofactor in many metabolic regulation reactions, into pyridoxine-deficient diets can increase Zn absorption [82]. Increasing the intake of milk or yogurt in plant-based diets has also been shown to increase the bioavailability of Zn [83]. Likewise, soybean extracts and one of its active components soyasaponin Bb have been found to increase ZIP4 importer and cellular Zn levels in mouse Hepa cells [84]. These observations regarding intestinal Zn bioavailability are promising and may be applicable to Zn bioavailability for the prostate as well. The plethora of modulating effects that these dietary phytochemicals possess on Zn levels as well as on Zn transporters creates the possibility that certain dietary phytochemicals alone or in combinations could be useful in the prevention or treatment of human PCa. Selected dietary phytochemicals studied in this reference (Table 1) are described below. 

### 9.1. Resveratrol

Resveratrol is one of the most commonly studied grape antioxidants for its health benefits and chemopreventive effects [94,95]. Studies show that antioxidants found in grapes may alleviate many health conditions including cancers [96,97,98]. Resveratrol’s effects against cancers including its ability to inhibit key signaling events associated with tumor development and progression, and multidrug resistance, make it is a unique natural agent in cancer management [99,100]. Recent studies on resveratrol alone or combination were shown to have promising anticancer effects against PCa [101,102,103]. However, most of these studies have not evaluated the effect of resveratrol on prostatic Zn levels. Interestingly, research has predicted that resveratrol molecules may form 1:1 (or up to 1.5 depending on Zn availability) complexes with Zn, facilitating Zn’s transport across cell membranes [85]. A study found that rodents had improved antioxidant capabilities along with increased concentrations of Zn in the plasma following resveratrol supplementation [86]. In an *in vitro* experimental model, resveratrol was shown to increase total cellular Zn levels in human prostate epithelial cells when also treated with Zn [85]. It also showed that resveratrol treatment significantly increased Zn concentration within the cell and induced cell cycle arrest at the G_2_/M phase of the cell cycle. The study demonstrated that resveratrol also aids in Zn supplementation’s effect of increasing free labile intracellular Zn, as well as its upregulation of metallothioneins. Finally, it found that resveratrol competed with zinquin for Zn *in vitro* [85]. Another study acknowledged resveratrol’s limited practical application due to its low solubility [87]. It suggested the use of resveratrol with metal ions in order to make it more soluble in polar solvents by increasing the molecule’s overall charge. Specifically, Al (III) and Zn (II) when in complexes with resveratrol decrease its oxidation potential, which is known to correlate to stronger antioxidant capabilities [87]. In summary, resveratrol can preserve Zn levels via the formation of Zn complexes, improve antioxidant activity and increase MTs expression; all these speak to resveratrol’s applications in Zn modulation for Zn homeostasis in PCa. These applications are significant and need to be further researched for their benefits.

### 9.2. Quercetin

Quercetin, a common flavonoid found in nuts, fruits and vegetables, has been found to display various anticancer properties including promoting apoptosis, autophagy, and decreased cell viability [104]. It has been shown that quercetin inhibits proliferation in multiple cancer cell lines including PCa. Several animal studies have also demonstrated anti-PCa effects of quercetin [103,105,106]. Quercetin has been found to induce apoptosis and necrosis by affecting mitochondrial integrity and homeostasis of reactive oxygen species (ROS) depending on the genetic makeup of PCa cells [107]. In PCa cells with mutated p53 and increased ROS, quercetin was found to reduce pro-survival protein kinase B (AKT) but activated proto-oncogenes RAF/ mitogen-activated protein kinase kinase (MEK) pathway. PCa cells lacking p53 and phosphatase and tensin homolog (PTEN) that also had lower ROS levels demonstrated activation of AKT and NF-κB pathways. Although these changes are known to be associated with both tumor suppressive and oncogenic responses, the cumulative effect was increased PCa cell death [107]. In addition, quercetin has been found to enhance the uptake of Zn in human intestinal Caco-2 cells [77,88]. A study found that quercetin is able to quickly increase the labile Zn in mouse hepatocarcinoma Hepa 1-6 cells [88]. Quercetin enhances the mobility of Zn via its ability to form a Zn-polyphenol chelation complex, which can move across the bilayer of liposomes [88]. In a very recent study from our lab, we evaluated the effect of quercetin and resveratrol combination in the diet in transgenic adenocarcinoma of the mouse prostate (TRAMP) and have found significant inhibition in PCa development and progression. This is a promising observation; however, this study did not evaluate to assess the level of Zn in response to quercetin and resveratrol combination [103]. These anti-cancer abilities of quercetin along with its ability to increase Zn uptake in the cells indicate that quercetin should continue to be studied for its applications in human PCa.

### 9.3. Epigallocatechin-3-Gallate

Epigallocatechin-3-gallate (EGCG), a major catechin in green tea, has been found to possess characteristics related to anti-proliferation, apoptosis induction, anti-metastasis, and tumor angiogenesis inhibition in various cancers while having no effect on benign cells [108,109]. A study found that high concentrations of EGCG and Zn^2+^ in LNCaP human PCa cell lines inhibit cell proliferation and decrease cell viability. Moreover, the treatment showed morphological changes in PCa cells: their membranes’ fluidity was decreased, and their growth was severely inhibited. EGCG was shown to increase the rate of total accumulation of Zn in the cytosol and mitochondria of PCa cells [89,90]. Its ability to increase the rate at which Zn accumulates in the cytosol and mitochondria of PCa cells makes it one of the promising dietary phytochemicals to prevent or treat PCa.

A different study found that EGCG, along with a grape seed procyanidin extract (GSPE) binds to Zn in solution with a higher affinity than zinquin. In the human hepatocarcinoma cell line HepG2, EGCG and GSPE were found to prevent Zn-induced toxicity [90]. Further, EGCG and GSPE inhibited the expression of metallothioneins and ZnT1, while simultaneously enhancing the expression of ZIP1 and ZIP4. GSPE also inhibited the expression of ZIP3, ZIP5, ZIP7, and ZIP11 while enhancing the expression of ZIP6, ZIP10, and ZIP13. Albumin and alpha-2-macroglobulin are the main carriers of Zn in plasma and are also both up-regulated by GSPE [90]. These transporters and carriers are significant in prostate health suggesting that EGCG and/or GSPE should be evaluated in studies for its Zn modulating effects for possible prevention and/or treatment of human PCa.

### 9.4. Curcumin

Curcumin a principal curcuminoid of turmeric has shown its beneficial effects against many cancers including PCa. Recently, in a placebo-controlled clinical trial evaluating the role of curcumin in PCa patients with intermittent androgen deprivation (IAD), though found no significant difference in the curve of off-treatment duration, PSA elevation was significantly suppressed [110]. Additionally, curcumin dose and duration used in this study (1440 mg/day for six months) were well tolerated and safe [110]. In another randomized, placebo-controlled study, curcumin has been found to increase total antioxidant capacity while decreasing superoxide dismutase, along with the reduction in PSA levels in PCa patients treated with radiotherapy [111]. These clinical studies along with several in vitro and in vivo studies show the clinical potential of curcumin against PCa; however, none of these studies evaluated the effect of curcumin on Zn levels in PCa tissues. Interestingly, a study has found that mice with benzo(a)pyrene-induced lung carcinogenesis that were treated with resveratrol and curcumin, were able to preserve Zn levels as well as regulate p21 protein’s function of inducing cell cycle arrest [91]. In the same study, it was shown that both mice treated with curcumin as well as mice treated with resveratrol individually had decreased incidences of tumorigenesis. When treated with both phytochemicals, the mice showed even fewer instances of tumors. Curcumin has also been found to improve Zn’s mobilization as well as to increase Zn-binding ligands and secure the host’s defense mechanisms, all of which support that curcumin increases the levels of intracellular Zn [91]. This suggests that curcumin may be a potential dietary phytochemical to modulate Zn homeostasis in PCa as well.

## 10. Agents Known to Inhibit the Absorption of Zn

There are certain agents that can inhibit the absorption of Zn (Table 1). Phytate, a compound found in many grains, nuts, seeds, and legumes, is a form of phosphorus used in the production of energy and the formation of cellular structures such as the membrane. Studies have found that phytic acid exhibits anticancer characteristics. These studies found phytate to be antineoplastic in various cells including leukemic hematopoietic K-562, colon cancer HT-29, breast cancer, cervical cancer, HepG2 hepatoma, and PCa cell lines, possibly affecting each cell line through different mechanisms. However, phytate has been found to inhibit Zn absorption because of its ability to form insoluble complexes with Zn in the presence of other cations such as Ca2+ and Mg2+, which then cannot be processed by the intestines [92,93]. Interestingly, in human dietary studies, it has been demonstrated that the calcium content does not increase the inhibitory effect of phytate for intestinal Zn absorption [112]. Dietary fibers have also been shown to form insoluble Zn complexes that cannot be processed by the intestines, resulting in lesser absorption of Zn [113]. Tin (element symbol Sn), a metal prevalent in canned foods, has also been found to inhibit Zn absorption. Study participants who consumed low amounts of tin were found to lose 0.7 mg of Zn through urine per day while participants who consumed higher amounts of tin were found to lose 0.6 mg of Zn through urine per day (reviewed in [92]). These are important observations and may help in decision-making when targeting Zn bioaccumulation in prostate malignancy.

## 11. Role of Zn in other Prostatic Diseases

Prostate disease is a prevalent cause of ailment today. Three common prostatic diseases are benign prostatic hyperplasia (BPH), prostatitis and PCa. In an analysis from 1000 candidates, prevalence rates of BPH, prostatitis and PCa are reported to be 23.1%, 5.1%, and 3.7% respectively [114]. BPH is a noncancerous increase in the size of the prostate. Hyperplasia of stromal and epithelial cells creates nodules in the prostate, and when large enough, blocks the urethra, causing urinary problems. An increase in age leads to an increase in aromatase and 5-alpha reductase, which convert androgen hormones to estrogen and dihydrotestosterone. Estrogen plays a role in the growth of cells in the prostate, which induces BPH. Prostatitis is the inflammation of the prostate because of a bacterial infection. All these three prostate diseases have been linked to the element Zn. Like prostatic carcinoma, prostate glands with BPH have shown to have a decrease in the concentration of tissue and plasma Zn, as well as an increase in urinary Zn when compared to the prostate gland of a normal healthy person [115]. Prostatitis is another prostate disease correlated to Zn levels. A study recently found that Zn treatment could potentially help with the management of this disease. It has been demonstrated that Zn blocks the effects of pro-inflammatory cytokines which contribute to chronic prostatitis. Using Zn with antibiotics has been shown to lower both the severity of the symptoms and the pain involved in chronic prostatitis and as well as to reduce the intra-urethral pressure compared to treatment with only antibiotics [116]. The role of Zn as a biomarker in prostatitis is another area of study currently being conducted. An experiment recently found that the Zn-binding proteins superoxide dismutase 3 (SOD3) and carbonic anhydrase 1 (CA1) were present in high quantities in prostatitis when compared to control [117]. These biomarkers could potentially be used when looking for new treatments for prostatitis in the future. These studies suggest that maintaining optimal Zn level could be a potential treatment strategy for these prostatic diseases as well.

## 12. Conclusions and Future Perspectives

Zn homeostasis is extremely important in human health because Zn is present in many structures of proteins such as Zn-dependent metalloenzymes and Zn-finger-containing transcriptional factors. Prostate cells accumulate a high amount of Zn to fulfill their unique metabolic needs. The importance of Zn homeostasis in PCa can be understood from the fact that there have been no confirmed cases of PCa in which Zn levels are not depleted [3]. In this review, we provided an overview of the current knowledge of Zn biology and its functions in the prostate. Multiple in vitro or in vivo studies suggest that the level of Zn in the system is modulated by naturally occurring dietary phytochemicals. Studies also suggest modulations in the regulators of Zn homeostasis in the cells including prostate cells in response to certain dietary phytochemicals. Thus, we conclude that Zn in combination with specific dietary phytochemicals may lead to enhanced intestinal Zn bioavailability as well as its uptake by prostate cells. It is expected that more bioavailable Zn will lead to enhanced bioaccumulation of Zn in the prostate cells and therefore, may inhibit PCa progression. Future perspectives may include the research on optimal Zn supplementation in combination with Zn modulating dietary phytochemicals, which may have superior anti-PCa effects, and detailed evaluation of involved mechanisms. Overall, it is expected that future in-depth studies with the ultimate goal of developing pharmacological modulators of Zn transporters and Zn homeostasis may prevent or reverse PCa.

## Figures and Tables

**Figure 1 nutrients-13-01867-f001:**
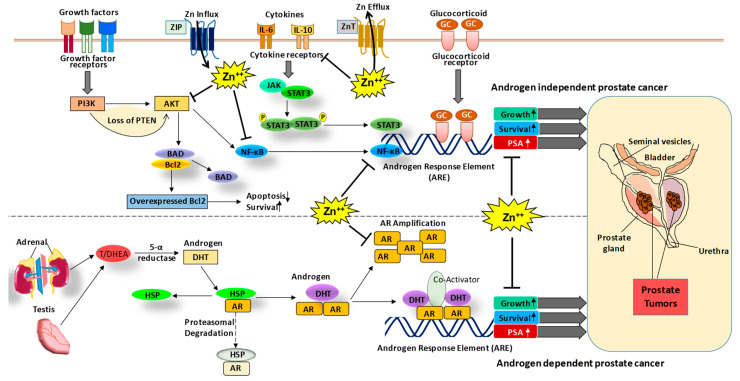
The role of Zn and associated signaling in PCa. The PCa can be progressed via androgen-dependent and independent signaling mechanisms. In the androgen-dependent pathway, testosterone hormone binds to the androgen receptor (AR), which leads to its dissociation from heat shock proteins (HSP) and enables its nuclear translocation and binding to the androgen response element (ARE) increasing the expression of growth and survival genes as well as PSA. In the androgen-independent mechanism, growth factor signaling, cytokines and glucocorticoid signaling activate cell proliferative pathways such as PI3K-AKT, NF-κB, and JAK-STAT, leading to an increase in expression of growth and survival genes as well as prostate-specific antigen (PSA) [17]. Interestingly, Zn is known to modulate AR and PSA [18] as well as other signaling molecules such as AKT [20] and NF-κB [21,22]. Thus, Zn modulating agents could be used in the prevention and/or treatment of PCa. DHT, dihydrotestosterone; T/ DHEA, testosterone/dehydroepiandrosterone.

**Table 1 nutrients-13-01867-t001:** **Dietary** phytochemicals known to modulate the levels of Zn.

Dietary Phytochemicals	Effect on Zn signaling	References
Green tea, red wine and grape juice	Increase Zn uptake Increase metallothioneins	[77]
Soybean extracts and soyasaponin Bb	Increase ZIP4 importer Increase cellular zinc levels	[84]
Resveratrol	Increase total cellular Zn levels Upregulate metallothioneins Form Zn complexes	[85,86,87]
Quercetin	Increase Zn uptake Form Zn-polyphenol chelation complex	[77,88]
Epigallocatechin-3-gallate (EGCG)	Increase accumulation of Zn Inhibit metallothioneins and ZnT1 Enhance ZIP1 and ZIP4	[89,90]
Grape-seed procyanidin extract (GSPE)	Inhibit ZIP3, ZIP5, ZIP7, and ZIP11 Enhance ZIP6, ZIP10, and ZIP13	[90]
Curcumin	Improve Zn’s mobilization Increase Zn-binding ligands	[91]
Phytate	Inhibit the absorption of Zn Form insoluble complexes with Zn	[92,93]

## Data Availability

Not applicable.

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
