# Peer review of "Dietary Phytochemicals in Zinc Homeostasis: A Strategy for Prostate Cancer Management"

_nutrients, 2021, doi:10.3390/nu13061867_

Round 1
Reviewer 1 Report
This manuscript is a really interesting review. The authors note four important highlights: First, the importance of zinc as an essential element in human biology and especially in prostate tissues. Second, zinc deficiency occurs in patients with prostate inflammation and prostate cancer. More significantly, the inverse correlation between zinc levels and cancer progression is observed. Third, although dietary changes and zinc supplementation can increase blood levels of zinc, this does not happen in the tissues of the prostate. The reason for this zinc deficiency in prostate tissues could be due to zinc transporters and regulators. Finally, the authors suggest that this zinc deficiency in the prostate can be recovered through natural dietary phytochemicals, especially quercetin, resveratrol, epigallocatechin-3-gallate, and curcumin. Suggesting the use of zinc and these phytochemicals as additional or adjunctive therapy against prostate cancer.
Even though the abstract is clear on the topic that the authors intend to convey, and the structure of the main points follows a clear, concise and concrete thought, the manuscript does not.
Major points:
The title should be more informative, it would be better “Dietary Phytochemicals May Increase Zinc Levels in Prostate Cancer: A review”
In the introduction, the authors should clearly state why this topic is important for this review. The main aim should be the same in the abstract (we have discussed the effect of selected phytochemicals -quercetin, resveratrol, epigallocatechin-3-gallate and curcumin-on Zn functioning in prostatic diseases and proposes that Zn in combination with specific dietary phytochemicals may lead to enhanced Zn bioaccumulation in the prostate, and therefore, may inhibit prostate cancer.) and introduction (we have discussed the role of Zn in prostate cancer and the effects of selected dietary phytochemicals modulating Zn levels to improve prostate cancer pathological conditions).
It would be better for the authors to present their hypothesis before the main objective of this review, which might be "the use of natural phytochemicals could increase the level of zinc in prostate tissues reducing the pathological conditions that increase the possibility of prostate cancer. They may be used as part of your treatment".
It would be better if the authors follow the same structure of the abstract. First, description of the normality of zinc in prostate tissues (why zinc is an essential micronutrient for the human body; the normal biochemistry of zinc -transporters, regulators, etc.-; the description of the singularities of zinc in tissues prostate cancer), how diet can improve or worsen zinc absorption and so on. Second, how, and why zinc levels change in inflammatory and malignant tissues of the prostate (zinc deficiency as a biomarker in prostate malignancy; epidemiology and racial disparity of prostate cancer, biochemical changes of zinc in abnormal tissues of the prostate, etc.). Finally, the reason for how and why natural dietary phytochemicals could increase zinc levels in prostate tissues using them as part of prostate cancer treatment. If the authors want to convey the importance of the use of phytochemicals as an important tool to prevent or combat prostate cancer, they need to improve their argumentation. It should be clear in a final paragraph (prior to conclusion) which natural phytochemicals described in this review are currently being used for this or other cancers and which ones should be studied in humans for their likely positive effects in animals or in vitro.
The conclusion should be their hypothesis supported by the argumentation of this manuscript.
Minor points:
Keywords should not be repeated if they appear in the title. There are a lot of abbreviations without their full name (table 1 and throughout the manuscript). Their full name must be entered when they first appear. The description in figure 1 should be improved to better understand what the authors are trying to report. What happens under normal conditions and what happens when zinc deficiency is present. Review the references throughout the manuscript.
Important information that authors should review is highlighted in the accompanying revised manuscript.
I would like to encourage the authors to rewrite this manuscript following a natural flow of knowledge on these important topics, zinc, prostate cancer and phytochemicals, going from the normal to the pathological, from the problem statement to a probable solution to the problem.

Author Response
Comments and Suggestions for Authors
This manuscript is a really interesting review. The authors note four important highlights: First, the importance of zinc as an essential element in human biology and especially in prostate tissues. Second, zinc deficiency occurs in patients with prostate inflammation and prostate cancer. More significantly, the inverse correlation between zinc levels and cancer progression is observed. Third, although dietary changes and zinc supplementation can increase blood levels of zinc, this does not happen in the tissues of the prostate. The reason for this zinc deficiency in prostate tissues could be due to zinc transporters and regulators. Finally, the authors suggest that this zinc deficiency in the prostate can be recovered through natural dietary phytochemicals, especially quercetin, resveratrol, epigallocatechin-3-gallate, and curcumin. Suggesting the use of zinc and these phytochemicals as additional or adjunctive therapy against prostate cancer. Even though the abstract is clear on the topic that the authors intend to convey, and the structure of the main points follows a clear, concise and concrete thought, the manuscript does not.
Response: We are very thankful to the reviewer for the appreciation of our work and careful reading of our manuscript. Our responses to the specific concerns are provided below.
Comment: The title should be more informative, it would be better “Dietary Phytochemicals May Increase Zinc Levels in Prostate Cancer: A review”
Response: We gratefully appreciate your comment. We have made a minor change in the title.
Comment: In the introduction, the authors should clearly state why this topic is important for this review. The main aim should be the same in the abstract (we have discussed the effect of selected phytochemicals -quercetin, resveratrol, epigallocatechin-3-gallate and curcumin-on Zn functioning in prostatic diseases and proposes that Zn in combination with specific dietary phytochemicals may lead to enhanced Zn bioaccumulation in the prostate, and therefore, may inhibit prostate cancer.) and introduction (we have discussed the role of Zn in prostate cancer and the effects of selected dietary phytochemicals modulating Zn levels to improve prostate cancer pathological conditions).
Response: We have revised the contents in the introduction section as suggested by the reviewer.
Comment: It would be better for the authors to present their hypothesis before the main objective of this review, which might be "the use of natural phytochemicals could increase the level of zinc in prostate tissues reducing the pathological conditions that increase the possibility of prostate cancer. They may be used as part of your treatment".
Response: We thank the reviewer for this comment. However, our review article was designed to be a narrative review, therefore, we did not present a hypothesis.
Comment: It would be better if the authors follow the same structure of the abstract. First, description of the normality of zinc in prostate tissues (why zinc is an essential micronutrient for the human body; the normal biochemistry of zinc -transporters, regulators, etc.-; the description of the singularities of zinc in tissues prostate cancer), how diet can improve or worsen zinc absorption and so on. Second, how, and why zinc levels change in inflammatory and malignant tissues of the prostate (zinc deficiency as a biomarker in prostate malignancy; epidemiology and racial disparity of prostate cancer, biochemical changes of zinc in abnormal tissues of the prostate, etc.). Finally, the reason for how and why natural dietary phytochemicals could increase zinc levels in prostate tissues using them as part of prostate cancer treatment. If the authors want to convey the importance of the use of phytochemicals as an important tool to prevent or combat prostate cancer, they need to improve their argumentation. It should be clear in a final paragraph (prior to conclusion) which natural phytochemicals described in this review are currently being used for this or other cancers and which ones should be studied in humans for their likely positive effects in animals or in vitro.
Response: We have organized some of the contents as the reviewer suggested to properly convey the importance of the use of phytochemicals for prostate cancer management.
Comment: The conclusion should be their hypothesis supported by the argumentation of this manuscript.
Response: The conclusion has been revised based on the reviewer’s above comments. However, we have avoided including a hypothesis-driven conclusion as this is not a hypothesis paper but a narrative review.
Comment: Keywords should not be repeated if they appear in the title. There are a lot of abbreviations without their full name (table 1 and throughout the manuscript). Their full name must be entered when they first appear. The description in figure 1 should be improved to better understand what the authors are trying to report. What happens under normal conditions and what happens when zinc deficiency is present. Review the references throughout the manuscript.
Response: We have changed the keywords as per the reviewer’s suggestion. We have also included the full name of genes when they first appear, and abbreviations thereafter. The description in Figure 1 has been updated. Based on the available literature about zinc modulation, Figure 1 is illustrated in androgen‐independent and dependent prostate cancer. Of course, this depicted information is in comparison to certain controls including normal conditions. We also re-checked the references.
Comment: Important information that authors should review is highlighted in the accompanying revised manuscript.
Response: We have revised the manuscript as per the suggestions.
Comment: I would like to encourage the authors to rewrite this manuscript following a natural flow of knowledge on these important topics, zinc, prostate cancer and phytochemicals, going from the normal to the pathological, from the problem statement to a probable solution to the problem.
Response: We have revised the manuscript as per the suggestion.
Reviewer 2 Report
This review aims to tackle an interesting topic: the role of phytochemicals on the overall zinc uptake into cancerous prostate and prostate development. Prostate cancer (PCa) is associated with a declined zinc accumulation in the epithelial cells of prostate peripheral tissue influencing the overall prostate function. Moreover, not only has the zinc status of PCa been found to be indicative of cancer status, but it is also discussed that increasing the zinc content in the prostate may positively influence this disease. As some phytochemicals are discussed to be beneficial for intestinal zinc bioavailability increasing zinc absorption, this review gives an overview on the current o far still very limited knowledge on these phytochemicals and their effect on intestinal zinc uptake in the intestine and impact on PCa. However, I still see a couple of problems with the manuscript as it is:
Major:
Line 56-50: “Studies have shown that oral ingestion of Zn is an inefficient way to supplement the required high Zn concentrations in the prostate due to malfunctioning of regulators of Zn homeostasis”. The motivation for reviewing the impact of dietary phytochemicals on body zinc status is not clearly explained. I think it would be helpful to briefly introduce the discussed beneficial role of these secondary plant products for zinc absorption in the intestine.
Line 52-68:The information in this section is interesting, but could be condensed further, particularly the inhibiting effect of zinc on prostate metabolism. Additionally, some parts are not clearly explained:
Line 54-58: What is the exact amount of this ‘extraordinary high’ zinc levels found in ‘healthy’ prostate fluid and peripheral prostate peripheral zone? Is there any information on intracellular zinc storage pools in prostate peripheral tissue? Is the zinc concentration in mitochondria of prostate cells higher than in other epithelial cells?
Line 58-62: Are there studies investigating the inhibition of mitochondrial aconitase by zinc (rates, affinity etc.)? If this has already been discussed in detail elsewhere, it could be useful to give a reference here.
Line 67-68: What is the consequence of the lower ATP production for prostate function?
Line 76-77: Why is the effect of zinc in prostate cells “unique”? Please elaborate this.
I would prefer more primary literature and less citation of other reviews, particularly when the cited information is an important part of the review and not a side information. For example, the references of the complete sections “Biochemistry of Zn in the healthy prostate” (lines 51-70) and “Zn and metabolic reprogramming in PCa” are predominantly reviews. Moreover, lines 136-142 and 145-149: The reference given here (Ref. 21) is a very good review, but it would improve the quality of this review to cite the original studies, for example from Andreini et al. 2005 for the statement lines 141-142 (Andreini, C., et al. (2005), Journal of Proteome Research 5(1): 196-201.). If a review is cited for the reason that it more comprehensively summarizes the information needed, it also helps to clearly identify it as such (for example by added “reviewed in X”).
Line 104-116: Figure 1:
It would be good if the underlying references on which Figure 1 is based were listed either directly in the figure or in the figure text to make it clear from which studies the individual pieces of information were derived.
The role of phytochemicals on zinc homeostasis is not clear from the representation in figure 1. Please make it clear in the illustration if they are upregulating zinc transporter expression or are increasing the uptake of zinc in the intestinal tract and thereby elevating zinc levels in blood circulation?
I think that the figure shows the cell membrane of the prostate epithelial cells and not that of the intestinal epithelium. The illustration as it is now implies that the phytochemicals act directly on the uptake into the prostate. However, the main in vitro studies performed so far have mainly investigated the effect of phytochemicals on uptake into enterocytes in vitro.
Line 132-133: “Since it is a trace element, its presence in the body depends on the dietary habits of individuals.” This is only partly true. There are also physiological factors, such as acrodermatitis enteropathica and gastrointestinal diseases (inflammatory bowel disease, diarrhea etc.) that impair zinc absorption from food in the intestinal tract (World Health Organization (2002) The World health report: 2002: Reducing risks, promoting healthy life.).
Line 136-138: “Zn is a fundamental component in the body’s defense mechanisms, as it is involved in mitosis, healing body wounds, and breakdown of carbohydrates.” I believe these are only some and very selected reasons for the essentiality of this micronutrient. Zinc is indispensable for the functioning of the immune system (Gammoh, N. Z. and L. Rink (2017). Nutrients 9(6).), which should definitely be mentioned here.
Line 162-164: “There is also a link between Zn concentrations in the prostate and levels of certain Zn transporters, which are important in the uptake and accumulation of Zn in prostate cells.” Please elaborate on this link in more detail. How are zinc transporters affected in PCa? Are their differentially expressed (upregulated/downregulated)?
Line 168-175: Cellular zinc homeostasis is tightly regulated by an elaborate system of zinc transporters, exporting, and sequestering cytoplasmic zinc (either in organelles or outside the cells) and by zinc binding proteins, such as metallothionein. This systems critically controls the cytoplasmic labile-bound zinc level, ensures that its concentration remains in the pM-nM range and prevents it from becoming toxic (Colvin, R.A et al. Metallomics 2010, 2, 306-317; https://doi.org/10.1039/b926662c). I do not understand the causal relationship between the expression of preventive mechanisms against cytoplasmic high zinc concentration, e.g. by the accumulation of zinc in organelles, and the conclusion that if these high zinc levels are absent, the preventive mechanisms must also be absent. Please explain this in more detail.
Line 175-179: What is the role of these microRNAs and transcription factors in preventing zinc cytotoxicity?
Line 245-246: Metallothionein are small proteins (~6kDa), not peptides (Coyle, P., et al. (2002). Cellular and Molecular Life Sciences 59(4): 627-647.)
Line 249: Please elaborate on what is known on the correlation of MT expression level and progression of PCa.
Line 251-252: “MT1 and MT2 search for and collect extracellular Zn but do not compete with essential Zn-requiring proteins” Even though the metallothionein isoforms MT 1 and 2 were found to be able to scavenge extracellular zinc (Palmiter, R.D, Toxicology and Applied Pharmacology 1995, 135, 139–146; https://doi.org/10.1006/taap.1995.1216), this is not their main purpose. These isoforms are still mainly cytoplasmic proteins that bind and buffer intracellular zinc and maintain stable cellular zinc homeostasis.
Line 260-261: “Approximately 90% of the Zn ions in the prostate are tightly bound to immobile macromolecules and ~10% are loosely bound to mobile ligands (about 10%).” Is it known which macromolecules are involved here? if so, it would certainly be good to give a few examples here, indicating the relevant literature used.
Line 266-267: “Further, Zn supplementation in humans has been found to increase metallothioneins, which control the bioavailability of Zn.” It is known that MT expression in the intestinal epithelium is upregulated as a response to elevated intestinal zinc absorption in vivo (Cragg, R.A. et al (2005) Gut, 54, 469-478; Richards, M.P.; Cousins, R.J. (1975) Biochemical and Biophysical Research Communications, 64, 1215-1223; Reeves, P.G. (1995). The Journal of Nutritional Biochemistry, 6, 48-54.). It also appears that elevated MT expression levels affect zinc absorption by the intestinal epithelial cells (Yasuno, T., et al. (2012). Biol Pharm Bull 35(4): 588-593; Davis, S.R.; McMahon, R.J.; Cousins, R.J. (1998) The Journal of Nutrition, 128, 825-831), which consequently leads to lower serum zinc levels. Yet there are many other cellular processes including the ZIP and ZnT protein family which are involved in controlling the cellular and systemic zinc bioavailability. Mostly the upregulation of ZnT-1 on the basolateral side of enterocytes, mediating the export of zinc from the enterocytes into the blood stream, are essential for the regulatory zinc absorption processes (Hennigar, S. R. and J. P. McClung (2018). Comprehensive Physiology 9(1): 59-74.). The statement that MT would take on this role alone shows only a small part of a complex system.
Line 258-260: This information would fit better to the section “ Role of Zn in the healthy prostate” (line 136) and is rather redundant at this point.
Lines 281-283, : “….even though Zn supplements increase the bioavailability of Zn in the body as a whole, they may not necessarily raise that of intra-prostatic Zn to desirable levels due to dysregulation in Zn transporters in PCa.” this sentence directly addresses one of the main problems of the review. The impaired transporter expression in PCa impairs zinc uptake into the prostate and is one of the main reasons why supplementation with other conventional zinc supplements does not appear to be effective in increasing zinc levels in the prostate. I wonder how the phytochemicals are supposed to work here? Most studies to date have investigated the effect of these phytochemicals on the intestinal uptake of zinc (mostly in in vitro studies). For example, in lines 353-354 you conclude from a study on the effect of phytochemicals on the intestinal uptake of zinc that these results may also apply to modulate Zn levels and prevent or treat PCa. Even if the intestinal bioavailability can be increased by these phytochemicals and as a result higher zinc plasma levels result than by the intake of other zinc components, how should the zinc reach the prostate? What is your hypothesis? Should the phytochemicals act directly on the uptake into the prostate or affect the dysregulated transporter expression in PCa? Please comment and explain this contradiction.
Line 297: As the authors of the review themselves point out in this section, the difference between Black Americans and white Americans probably has more to do with epigenetic factors than with race, hence I would suggest changing the title of this section into simply: “Zn deficiency and PCa epidemiology”.
Line 299: Please elaborate what is meant with an “subpar zinc ingestion”. Is oral absorption or intestinal absorption meant here and what are the consequences?
Lines 340-369: I would strongly recommend that the two different uses of the term zinc bioavailability be defined and used accordingly throughout the manuscript to avoid misunderstandings: There is zinc bioavailability in the intestine and systemic zinc bioavailability, e.g. for body functions such as availability for prostate or absorption into the prostate. These are two different topics which are either regulated by the food matrix/ food components that positively influence the intestinal zinc availability and its uptake by enterocytes or by systemic factors. Please refer to intestinal zinc bioavailability and systemic zinc bioavailability or availability for prostate.
Lines 370-465: Can you elaborate on the concentrations of phytochemicals used in the individual in vitro studies and compare them with physiologically relevant concentrations (in the human plasma upon their absorption) and daily tolerable intakes of these secondary plant compounds? This would certainly be interesting and would more clearly highlight the relevance of the results of the individual studies and the applicability for the treatment of PCa.
Line 400: Please also list the references for each study in the table.
Lines 403-412: Please elaborate on the consequences of the affected Akt- and NfkB-pathways in PCa by quercetin.
Line 475: Even though the zinc-phytate-these complexes were found to be stronger in the presence of calcium (Fredlund, K.; Isaksson, M.; Rossander-Hulthén, L.; Almgren, A.; Sandberg, A.-S. (2006), Journal of Trace Elements in Medicine and Biology, 20, 49-57), several human dietary studies demonstrated that the calcium content does not increase the inhibitory effect of phytate for the intestinal zinc absorption (Sandström, Bet al. (1980) The American Journal of Clinical Nutrition 1980, 33, 739-745; Lönnerdal, B.et al. (1984) The American Journal of Clinical Nutrition, 40, 1064-1070; Hunt, J.R.; Beiseigel, J.M. (2009) The American Journal of Clinical Nutrition, 89, 839-843.).
Line 476: What exactly do you mean by “certain” fibers? please be more specific.
Minor:
Line 13: “correlate”
Line 20: “discuss”
Line 72: ..”the” Warburg effect
Line 202: Please extend the abbreviations of ZIP and ZnT with Zrt-, Irt-like protein (ZIP)4 (solute carrier(SLC)39)) and ZnT (zinc transporter).
Line 231: Please rephrase the sentence to “….enter the cell”.
Line: 427: Please introduce the cell line LNCaP briefly. Is this a human cell line and what are its advantages?
Author Response
Response to Comments of Reviewer #2
Comments and Suggestions for Authors
This review aims to tackle an interesting topic: the role of phytochemicals on the overall zinc uptake into cancerous prostate and prostate development. Prostate cancer (PCa) is associated with a declined zinc accumulation in the epithelial cells of prostate peripheral tissue influencing the overall prostate function. Moreover, not only has the zinc status of PCa been found to be indicative of cancer status, but it is also discussed that increasing the zinc content in the prostate may positively influence this disease. As some phytochemicals are discussed to be beneficial for intestinal zinc bioavailability increasing zinc absorption, this review gives an overview on the current o far still very limited knowledge on these phytochemicals and their effect on intestinal zinc uptake in the intestine and impact on PCa. However, I still see a couple of problems with the manuscript as it is:
Response: The authors would like to thank the reviewer for the positive comments. Our responses to the specific concerns are provided below.
Comment: Line 46-50: “Studies have shown that oral ingestion of Zn is an inefficient way to supplement the required high Zn concentrations in the prostate due to malfunctioning of regulators of Zn homeostasis”. The motivation for reviewing the impact of dietary phytochemicals on body zinc status is not clearly explained. I think it would be helpful to briefly introduce the discussed beneficial role of these secondary plant products for zinc absorption in the intestine.
Response: We have revised this section and included how certain phytochemicals affect zinc levels by modulating zinc transporters and their regulators.
Comment: Line 52-68:The information in this section is interesting, but could be condensed further, particularly the inhibiting effect of zinc on prostate metabolism. Additionally, some parts are not clearly explained:
Response: We revised and clarified this section.
Comment: Line 54-58: What is the exact amount of this ‘extraordinary high’ zinc levels found in ‘healthy’ prostate fluid and peripheral prostate peripheral zone? Is there any information on intracellular zinc storage pools in prostate peripheral tissue? Is the zinc concentration in mitochondria of prostate cells higher than in other epithelial cells?
Response: We have added the known amount of zinc levels in the prostatic fluid. All other information are included in section 2. Biochemistry of Zn in the healthy prostate’.
Comment: Line 58-62: Are there studies investigating the inhibition of mitochondrial aconitase by zinc (rates, affinity etc.)? If this has already been discussed in detail elsewhere, it could be useful to give a reference here.
Response: We have provided additional references.
Comment: Line 67-68: What is the consequence of the lower ATP production for prostate function?
Response: Low ATP production in the prostate is caused by the truncation of the Krebs cycle at the first step of citrate oxidation, which also generates citrate as the final product. Citrate is an important constituent of semen and high citrate levels in the prostatic fluid are required for several prostatic functions.
Comment: Line 76-77: Why is the effect of zinc in prostate cells “unique”? Please elaborate this. I would prefer more primary literature and less citation of other reviews, particularly when the cited information is an important part of the review and not a side information. For example, the references of the complete sections “Biochemistry of Zn in the healthy prostate” (lines 51-70) and “Zn and metabolic reprogramming in PCa” are predominantly reviews. Moreover, lines 136-142 and 145-149: The reference given here (Ref. 21) is a very good review, but it would improve the quality of this review to cite the original studies, for example from Andreini et al. 2005 for the statement lines 141-142 (Andreini, C., et al. (2005), Journal of Proteome Research 5(1): 196-201.). If a review is cited for reason that it more comprehensively summarizes the information needed, it also helps to clearly identify it as such (for example by added “reviewed in X”).
Response: Prostate cells accumulate a high amount of zinc to inhibit mitochondrial aconitase resulting truncation of Krebs cycle at the first step of citrate oxidation. Due to this unique metabolic effect, healthy prostate cells avoid oxidative phosphorylation and generate only 14 ATP per glucose molecule instead of the typical 38 ATP from the complete oxidation of glucose. We have elaborated on this portion and provided more literature to support this fact.
Comment: Line 104-116: Figure 1: It would be good if the underlying references on which Figure 1 is based were listed either directly in the figure or in the figure text to make it clear from which studies the individual pieces of information were derived.
Response: We have included the required references in the legend of figure 1.
Comment: The role of phytochemicals on zinc homeostasis is not clear from the representation in figure 1. Please make it clear in the illustration if they are upregulating zinc transporter expression or are increasing the uptake of zinc in the intestinal tract and thereby elevating zinc levels in blood circulation?
Response: As the role of phytochemicals on zinc homeostasis is already detailed in Table 1, we have limited Figure 1 to schematically represent the role of Zn and associated signaling in prostate cancer. To avoid any confusion, we have removed phytochemicals from Figure 1.
Comment: I think that the figure shows the cell membrane of the prostate epithelial cells and not that of the intestinal epithelium. The illustration as it is now implies that the phytochemicals act directly on the uptake into the prostate. However, the main in vitro studies performed so far have mainly investigated the effect of phytochemicals on uptake into enterocytes in vitro.
Response: We agree with the reviewer that most of the data derived for Figure 1 are from in vitro studies. For clarity and to avoid any confusion, we have removed phytochemicals and revised Figure 1. As limited studies are available on the prostate, our focus has been whether dietary phytochemicals modulate zinc levels in any of the in vitro or in vivo systems. It is expected that more bioavailable zinc will lead to enhanced bioaccumulation of Zn in prostate cells as well.
Comment: Line 132-133: “Since it is a trace element, its presence in the body depends on the dietary habits of individuals.” This is only partly true. There are also physiological factors, such as acrodermatitis enteropathica and gastrointestinal diseases (inflammatory bowel disease, diarrhea etc.) that impair zinc absorption from food in the intestinal tract (World Health Organization (2002) The World health report: 2002: Reducing risks, promoting healthy life.).
Response: Thank you. We have included ‘physiological factors’ in the sentence.
Comment: Line 136-138: “Zn is a fundamental component in the body’s defense mechanisms, as it is involved in mitosis, healing body wounds, and breakdown of carbohydrates.” I believe these are only some and very selected reasons for the essentiality of this micronutrient. Zinc is indispensable for the functioning of the immune system (Gammoh, N. Z. and L. Rink (2017). Nutrients 9(6).), which should definitely be mentioned here.
Response: As suggested, we have included the role of zinc in the functioning of the immune system and added the reference Gammoh, N. Z. and L. Rink (2017).
Comment: Line 162-164: “There is also a link between Zn concentrations in the prostate and levels of certain Zn transporters, which are important in the uptake and accumulation of Zn in prostate cells.” Please elaborate on this link in more detail. How are zinc transporters affected in PCa? Are their differentially expressed (upregulated/downregulated)?
Response: This has been discussed in a separate section 6. “Important regulators of Zn homeostasis in the prostate”.
Comment: Line 168-175: Cellular zinc homeostasis is tightly regulated by an elaborate system of zinc transporters, exporting, and sequestering cytoplasmic zinc (either in organelles or outside the cells) and by zinc binding proteins, such as metallothionein. This systems critically controls the cytoplasmic labile-bound zinc level, ensures that its concentration remains in the pM-nM range and prevents it from becoming toxic (Colvin, R.A et al. Metallomics 2010, 2, 306-317; https://doi.org/10.1039/b926662c). I do not understand the causal relationship between the expression of preventive mechanisms against cytoplasmic high zinc concentration, e.g. by the accumulation of zinc in organelles, and the conclusion that if these high zinc levels are absent, the preventive mechanisms must also be absent. Please explain this in more detail.
Response: Colvin, R.A et al. have not discussed the role of zinc in the prostate. However, we have revised this section to make it clear.
Comment: Line 175-179: What is the role of these microRNAs and transcription factors in preventing zinc cytotoxicity?
Response: These microRNAs and transcription factors play key roles in the regulation of zinc levels. We have revised this portion as suggested.
Comment: Line 245-246: Metallothionein are small proteins (~6kDa), not peptides (Coyle, P., et al. (2002). Cellular and Molecular Life Sciences 59(4): 627-647.)
Response: We have corrected this and cited the reference Coyle, P., et al. (2002).
Comment: Line 249: Please elaborate on what is known on the correlation of MT expression level and progression of PCa.
Response: This has been discussed in the last paragraph of section 6. “Important regulators of Zn homeostasis in the prostate”.
Comment: Line 251-252: “MT1 and MT2 search for and collect extracellular Zn but do not compete with essential Zn-requiring proteins” Even though the metallothionein isoforms MT 1 and 2 were found to be able to scavenge extracellular zinc (Palmiter, R.D, Toxicology and Applied Pharmacology 1995, 135, 139–146; https://doi.org/10.1006/taap.1995.1216), this is not their main purpose. These isoforms are still mainly cytoplasmic proteins that bind and buffer intracellular zinc and maintain stable cellular zinc homeostasis.
Response: We agree, and this is detailed at the beginning of the paragraph.
Comment: Line 260-261: “Approximately 90% of the Zn ions in the prostate are tightly bound to immobile macromolecules and ~10% are loosely bound to mobile ligands (about 10%).” Is it known which macromolecules are involved here? if so, it would certainly be good to give a few examples here, indicating the relevant literature used.
Response: Immobile Zn ions are mostly protein-bound. We have revised this sentence in the main text.
Comment: Line 266-267: “Further, Zn supplementation in humans has been found to increase metallothioneins, which control the bioavailability of Zn.” It is known that MT expression in the intestinal epithelium is upregulated as a response to elevated intestinal zinc absorption in vivo (Cragg, R.A. et al (2005) Gut, 54, 469-478; Richards, M.P.; Cousins, R.J. (1975) Biochemical and Biophysical Research Communications, 64, 1215-1223; Reeves, P.G. (1995). The Journal of Nutritional Biochemistry, 6, 48-54.). It also appears that elevated MT expression levels affect zinc absorption by the intestinal epithelial cells (Yasuno, T., et al. (2012). Biol Pharm Bull 35(4): 588-593; Davis, S.R.; McMahon, R.J.; Cousins, R.J. (1998) The Journal of Nutrition, 128, 825-831), which consequently leads to lower serum zinc levels. Yet there are many other cellular processes including the ZIP and ZnT protein family which are involved in controlling the cellular and systemic zinc bioavailability. Mostly the upregulation of ZnT-1 on the basolateral side of enterocytes, mediating the export of zinc from the enterocytes into the blood stream, are essential for the regulatory zinc absorption processes (Hennigar, S. R. and J. P. McClung (2018). Comprehensive Physiology 9(1): 59-74.). The statement that MT would take on this role alone shows only a small part of a complex system.
Response: We have reworded this sentence in the revised manuscript. ZIP and ZnT protein families are discussed in section 6. “Important regulators of Zn homeostasis in the prostate”.
Comment: Line 258-260: This information would fit better to the section “ Role of Zn in the healthy prostate” (line 136) and is rather redundant at this point.
Response: We have moved the sentence as suggested.
Comment: Lines 281-283, : “….even though Zn supplements increase the bioavailability of Zn in the body as a whole, they may not necessarily raise that of intra-prostatic Zn to desirable levels due to dysregulation in Zn transporters in PCa.” this sentence directly addresses one of the main problems of the review. The impaired transporter expression in PCa impairs zinc uptake into the prostate and is one of the main reasons why supplementation with other conventional zinc supplements does not appear to be effective in increasing zinc levels in the prostate. I wonder how the phytochemicals are supposed to work here? Most studies to date have investigated the effect of these phytochemicals on the intestinal uptake of zinc (mostly in in vitro studies). For example, in lines 353-354 you conclude from a study on the effect of phytochemicals on the intestinal uptake of zinc that these results may also apply to modulate Zn levels and prevent or treat PCa. Even if the intestinal bioavailability can be increased by these phytochemicals and as a result higher zinc plasma levels result than by the intake of other zinc components, how should the zinc reach the prostate? What is your hypothesis? Should the phytochemicals act directly on the uptake into the prostate or affect the dysregulated transporter expression in PCa? Please comment and explain this contradiction.
Response: This concern has been elaborated in “Section 9. Naturally occurring dietary phytochemicals known to enhance the absorption/bioaccumulation of Zn” and summarized in Table 1. Yes, there are several studies available showing the effect of these phytochemicals on the dysregulated zinc transporters (Table 1).
Comment: Line 297: As the authors of the review themselves point out in this section, the difference between Black Americans and white Americans probably has more to do with epigenetic factors than with race, hence I would suggest changing the title of this section into simply: “Zn deficiency and PCa epidemiology”.
Response: We have changed the title of this section to “Zn deficiency and PCa epidemiology” as suggested.
Comment: Line 299: Please elaborate what is meant with an “subpar zinc ingestion”. Is oral absorption or intestinal absorption meant here and what are the consequences?
Response: It referred to the total zinc consumption from all the sources. We have revised this sentence to make it clear.
Comment: Lines 340-369: I would strongly recommend that the two different uses of the term zinc bioavailability be defined and used accordingly throughout the manuscript to avoid misunderstandings: There is zinc bioavailability in the intestine and systemic zinc bioavailability, e.g. for body functions such as availability for prostate or absorption into the prostate. These are two different topics which are either regulated by the food matrix/ food components that positively influence the intestinal zinc availability and its uptake by enterocytes or by systemic factors. Please refer to intestinal zinc bioavailability and systemic zinc bioavailability or availability for prostate.
Response: Thank you for this suggestion. We have revised the manuscript accordingly.
Comment: Lines 370-465: Can you elaborate on the concentrations of phytochemicals used in the individual in vitro studies and compare them with physiologically relevant concentrations (in the human plasma upon their absorption) and daily tolerable intakes of these secondary plant compounds? This would certainly be interesting and would more clearly highlight the relevance of the results of the individual studies and the applicability for the treatment of PCa.
Response: Yes, it would be very informative if one could compare the in vitro doses to human exposure. However, this will be challenging because of the variable parameters used in in vitro studies and the complex nature of clinical studies.
Comment: Line 400: Please also list the references for each study in the table.
Response: We have added references for each study in Table 1.
Comment: Lines 403-412: Please elaborate on the consequences of the affected Akt- and NfkB-pathways in PCa by quercetin.
Response: As suggested, we have elaborated this section to include the consequences of the affected AKT- and NF-kB-pathways in PCa by quercetin.
Comment: Line 475: Even though the zinc-phytate-these complexes were found to be stronger in the presence of calcium (Fredlund, K.; Isaksson, M.; Rossander-Hulthén, L.; Almgren, A.; Sandberg, A.-S. (2006), Journal of Trace Elements in Medicine and Biology, 20, 49-57), several human dietary studies demonstrated that the calcium content does not increase the inhibitory effect of phytate for the intestinal zinc absorption (Sandström, Bet al. (1980) The American Journal of Clinical Nutrition 1980, 33, 739-745; Lönnerdal, B.et al. (1984) The American Journal of Clinical Nutrition, 40, 1064-1070; Hunt, J.R.; Beiseigel, J.M. (2009) The American Journal of Clinical Nutrition, 89, 839-843.).
Response: Thank you for pointing this out. We have revised the text accordingly.
Comment: Line 476: What exactly do you mean by “certain” fibers? Please be more specific.
Response: Thank you. We have revised this sentence and referenced it accordingly.
Comment: Line 13: “correlate” Line 20: “discuss” Line 72: ..”the” Warburg effect
Response: We have added” the” in line 72, however, we are not seeing any problem with lines 13 and 20.
Comment: Line 202: Please extend the abbreviations of ZIP and ZnT with Zrt-, Irt-like protein (ZIP)4 (solute carrier(SLC)39)) and ZnT (zinc transporter).
Response: We have included the full name of genes when they first appear, and abbreviations thereafter.
Comment: Line 231: Please rephrase the sentence to “….enter the cell”.
Response: Thank you. We have rephrased the sentence as suggested.
Comment: Line: 427: Please introduce the cell line LNCaP briefly. Is this a human cell line and what are its advantages?
Response: This is a human prostate cancer cell line. We have updated the text.
Round 2
Reviewer 1 Report
The manuscript has improved significantly, small changes are suggested before publication. The modifications made (highlighted in green) made the manuscript more understandable and conveyed what was proposed by the authors. There have been significant improvements in the introduction, table 1 and conclusions. The abbreviations in which the full name must be written have been highlighted in red.

Reviewer 2 Report
Thank you for the opportunity to read this resubmission and the detailed response to my comments. I believe the authors have done a good job at improving the manuscript and it has benefitted greatly from the revision.